# Physiological Parameters and Transcriptomic Levels Reveal the Response Mechanism of Maize to Deep Sowing and the Mechanism of Exogenous MeJA to Alleviate Deep Sowing Stress

**DOI:** 10.3390/ijms251910718

**Published:** 2024-10-05

**Authors:** Fang Wang, Zhijin Feng, Xinyi Yang, Guangkuo Zhou, Yunling Peng

**Affiliations:** 1College of Agronomy, Gansu Agricultural University, Lanzhou 730070, China; wangfang@gsau.edu.cn (F.W.); f15763075067@163.com (Z.F.); yxy11102024@163.com (X.Y.); m17899314410@163.com (G.Z.); 2Gansu Provincial Key Laboratory of Aridland Crop Science, Gansu Agricultural University, Lanzhou 730070, China

**Keywords:** maize, exogenous methyl jasmonate, deep-sowing stress, mitigation effect

## Abstract

Deep sowing, as a method to mitigate drought and preserve soil moisture and seedlings, can effectively mitigate the adverse effects of drought stress on seedling growth. The elongation of the hypocotyl plays an important role in the emergence of maize seeds from deep-sowing stress. This study was designed to explore the function of exogenous methyl jasmonate (MeJA) in the growth of the maize mesocotyl and to examine its regulatory network. The results showed that the addition of a 1.5 μ mol L^−1^ MeJA treatment significantly increased the mesocotyl length (MES), mesocotyl and coleoptile length (MESCOL), and seedling length (SDL) of maize seedlings. Transcriptome analysis showed that exogenous MeJA can alleviate maize deep-sowing stress, and the differentially expressed genes (DEGs) mainly include ornithine decarboxylase, terpene synthase 7, ethylene responsive transcription factor 11, and so on. In addition, candidate genes that may regulate the length of maize hypocotyls were screened by Weighted Gene Co-expression Network Analysis (WGCNA). These genes may be involved in the growth of maize hypocotyls through transcriptional regulation, histones, ubiquitin protease, protein binding, and chlorophyll biosynthesis and play an important role in maize deep-sowing tolerance. Our research findings may provide a theoretical basis for determining the tolerance of maize to deep-sowing stress and the mechanism of exogenous hormone regulation of deep-sowing stress.

## 1. Introduction

Maize (*Zea mays* L.) is an important crop for both food and feed. It is very important to ensure high and stable yields of maize for food production safety. Drought and water shortage are some of the most important abiotic stresses that cause global maize yield reductions. Most of the maize planting areas in China are located in arid and semi-arid areas. According to statistics, the annual yield reduction caused by drought and water shortage is about 20–50% [1,2]. In recent years, in order to solve the problem of grain yield reduction caused by drought, many scholars have proposed different solutions, mainly including cultivating new drought-tolerant varieties [3] and deep-sowing seeds [4]. However, because the breeding of new drought-tolerant varieties is related to the genetic mechanism and drought tolerance is controlled by a variety of micro-effect genes, its regulation mechanism is still unclear and it is difficult to apply such methods on a large scale. Therefore, it is relatively simple and feasible to reduce drought stress by deep sowing of maize seeds. At the same time, deep sowing is also an efficient measure for drought resistance and seedling protection in maize and a new way of cultivating drought-resistant maize [5].

At present, the research on the deep-sowing tolerance of maize mainly focuses on the selection of deep-sowing quality and characteristics, while the research on the role of exogenous hormones in alleviating deep-sowing stress in the regulation of the hypocotyl is less advanced, and the research results are not the same, so a systematic evaluation system cannot be formed. Studies have shown that with increasing sowing depth, the main reason for maize tolerance to deep sowing is the significant elongation of the mesocotyl, and the change in mesocotyl length is closely related to hormone regulation [6]. Suge et al. [7] found that gibberellin, auxin, abscisic acid, and ethylene all promote the elongation of rice hypocotyls. ABA mainly increases the number of mesodermal axis cells through cell analysis; Kiyoshi’s [8] study showed that simultaneous application of exogenous ETH and GA can significantly elongate the length of the hypocotyl in indica rice. Watanabe et al. [9] found that GA, ABA, IAA, and ethylene can all promote the elongation of rice hypocotyls. ABA promotes cell division, while GA, IAA, and ethylene mainly promote cell elongation. Although there have been a lot of transcriptome studies on maize tolerance to deep sowing, there are few reports on the mechanism of exogenous hormones regulating mesocotyl growth in response to deep-sowing stress.

Methyl jasmonate (MeJA), a hormone that responds to plant damage, is widely used for plant resistance to damage stress [10]. Studies have shown that MeJA, as an internal signal transduction molecule, has the characteristics of easy transport, non-ionization, and easy membrane penetration [11]. However, the current research on alleviating deep-sowing stress by exogenous MeJA has not been reported, and the mechanism of alleviating deep-sowing stress is still unclear. Therefore, we assume that exogenous application of MeJA can effectively alleviate the deep-sowing stress of maize, promote the growth of the maize seedling hypocotyl, and improve its drought resistance. In this study, the deep-sowing-tolerant maize inbred line Qi319 and the deep-sowing-sensitive maize inbred line Zi330 were used as materials. The differences in the mesocotyls of the different inbred lines under deep sowing and exogenous MeJA treatment were studied in aspects of phenotype, cell structure, physiology, and biochemistry, and the metabolic regulatory networks and regulatory genes related to deep sowing were studied via transcriptomics. The aim of the study was to reveal the molecular mechanisms of deep-sowing-tolerance-related traits in maize and lay a foundation for the exploration of deep-sowing-tolerance genes.

## 2. Results

### 2.1. Physiological Characteristics of Exogenous MeJA in Alleviating Deep-Sowing Stress in Maize Inbred Lines

#### 2.1.1. Morphological Analysis of Exogenous MeJA Alleviating Deep-Sowing Stress in Maize Inbred Lines

At a sowing depth of 15 cm, the growth of the two maize inbred lines was significantly affected. After the addition of the MeJA treatment, the material accumulation of the seedlings of both inbred lines increased in the DM treatment. The seedling length (SDL), root length (RL), seedling fresh weight (SDW), and root fresh weight (RW) of the deep-sowing inbred line Qi319 were significantly increased by 34.26%, 13.97%, 7.48%, and 17.54%, respectively. The seedling length (SDL), seedling fresh weight (SDW), and root fresh weight (RW) of the deep-sowing-sensitive inbred lines of Zi330 increased by 8.96%, 7.69%, and 33.33%, respectively. The results showed that MeJA was the main factor in the enhancement of seed stress resistance under deep sowing (Table 1).

Compared with DS, the length of the hypocotyls of the deep-sowing-sensitive inbred line Zi330 increased by 26.51%, while that of the deep-sowing-tolerant inbred line Qi319 increased by 7.5%. It can be seen that MeJA has different effects in terms of the improvement of deep-sowing tolerance in different varieties, indicating that the deep sowing of seeds is regulated by both the genetic characteristics of varieties and the exogenous hormone MeJA. The application of the exogenous hormone can adjust the deep-sowing tolerance of maize seeds (Table 2).

#### 2.1.2. Endogenous Hormone Analysis of Exogenous MeJA Alleviating Deep-Sowing Stress in Maize Inbred Lines

It can be seen from Figure 1 that after the addition of exogenous MeJA, the endogenous hormones of Zi330 and Q319 changed significantly, and the endogenous hormone changes were particularly significant under deep-sowing conditions. Compared to the DS treatment, the levels of growth hormone (IAA) and gibberellin (GA_3_) in the mesocotyls of the deep-sowing-sensitive Zi330 inbred lines increased significantly under the DM treatment, suggesting that MeJA may have a synergistic effect with IAA and GA_3_, while abscisic acid (ABA) and brasinolide (EBR) contents decreased significantly in response to deep-sowing stress. It is suggested that MeJA may have an antagonistic relation to ABA and EBR. At the sowing depths of 3 cm and 15 cm, the endogenous MeJA content in the mesocotyls of the two maize inbred lines decreased to different degrees after MeJA was applied, which may have been due to the low content of the optimal hormone for MeJA response to deep sowing of maize. It is speculated that when exogenous MeJA is applied, the content of endogenous MeJA in the mesocotyls of the two maize inbred lines decreases to different degrees. Endogenous MeJA produced by plant mesocotyls can alleviate the damage caused by deep-sowing stress. The results indicated that endogenous hormones could stabilize the internal environment balance and promote the growth and development of plants by regulating their own levels under deep-sowing conditions.

#### 2.1.3. Cytological Observation of Exogenous MeJA Alleviating Deep-Sowing Stress in Maize Inbred Lines

As shown in Figure 2, the mesocotyls of the two maize inbred line seedlings at the 15 cm sowing depth increased significantly compared with the 3 cm sowing depth, indicating that the response of maize seedlings to deep-sowing stress was closely related to mesocotyl length. The length of mesocotyl cells in the two inbred lines increased after the addition of the 1.5 μmol·L^−1^ exogenous MeJA treatment at the sowing depths of 3 cm and 15 cm, and the length of mesocotyl cells increased most significantly under the DM treatment. Compared with the CK treatment, the mesocotyl cell lengths of Zi330 and Qi319 increased by 25.98% and 19.69% under the CM treatment. Compared with DS, the mesocotyl cell length of Zi330 and Qi319 increased by 46.44% and 31.99% under the DM treatment.

### 2.2. Transcriptome Analysis of Exogenous MeJA Alleviating Deep-Sowing Stress

#### 2.2.1. Analysis of Sequencing Results

The constructed library was sequenced with the DNBSEQ platform for transcriptomes; 24 samples were sequenced, and a total of 31,384 genes were detected. Each sample yielded an average of 45.64 M original reads. After filtering and quality control of the original reads, each sample yielded an average of approximately 44.18 M valid data, with a Q20% value range of 96.41–96.65%. The amount of sample sequencing data is shown in Appendix A.

#### 2.2.2. Identification and Functional Analysis of Differentially Expressed Genes

In order to comprehensively study the deep-sowing tolerance of maize and the mitigation effect of exogenous MeJA on deep-sowing stress in maize, we used FPKM to calculate gene expression, and genes with a differencing multiple of gene expression |log_2_FC| ≥ 1 and *p* ≤ 0.05 were defined as DEGs. As can be seen in Figure 3a, compared with the normal sowing depth, 1248 and 1288 DEGs were identified under deep-sowing stress, accounting for 48.79% and 40.06% of the total differential genes, respectively, while 1310 and 1927 genes were down-regulated, respectively. They accounted for 51.21% and 59.94% of the total differential genes, respectively. Compared with the addition of exogenous MeJA under the normal sowing depth, 538 DEGs were up-regulated and 94 DEGs were down-regulated after exogenous MeJA was applied to Qi319 under deep-sowing stress. However, 2284 DEGs were up-regulated and 3601 DEGs were down-regulated after the addition of exogenous MeJA under deep-sowing stress, indicating that the deep-sowing tolerance of Zi330 was enhanced by more down-regulated DEGs under exogenous MeJA. Similarly, analysis of the distribution of common and specific differential genes in a Venn diagram also fully demonstrated the mitigating effect of the exogenous hormone MeJA on maize inbred lines with different deep-sowing-tolerance characteristics. As shown in Figure 3b, 225 different genes were co-expressed between the two inbred lines under deep-sowing stress compared with the normal sowing depth. There were 42 differential genes that were co-expressed after exogenous MeJA was applied under deep-sowing stress compared with the normal sowing depth. With distilled water applied at a normal sowing depth as a control, there were six differential genes co-expressed after exogenous MeJA was applied. In contrast, with the application of distilled water at a depth of 15 cm, there were three differential genes that were co-expressed after the application of exogenous MeJA (Figure 3c, Appendix A).

#### 2.2.3. GO Analysis of Differentially Expressed Genes

GO functional annotations can be divided into three main categories: biological processes (BPs), cellular components (CCs), and molecular functions (MFs). GO functional analysis of DEGs in the two maize inbred lines subjected to deep-sowing stress was conducted. Under deep-sowing stress, the main BPs of DEG enrichment in the deep-sown inbred line Qi319 were cellular processes, followed by metabolic processes and biological regulation. The main enriched CC was the cell, followed by cell parts, organelles and membranes. The main enriched MFs were catalytic activity, binding, and transporter activity. The terms with significant enrichment of DEGs in BPs and CCs of the deep-sowing-sensitive inbred lines of Zi330 were basically the same as those of Qi319. The main enriched MFs were binding, catalytic activity, and structural molecule activity, which was different from Qi319. It can be seen that the DEGs identified in this study mainly relate to metabolic regulation, cell metabolism, active molecule binding, and transport (Figure 4a,b).

GO functional annotation analysis was carried out on the DEGs identified in the two inbred lines under the treatment of exogenous MeJA to alleviate deep-sowing stress (Figure 4c,d). Biological processes, cellular components, and molecular functions were the main enrichment processes of differentially expressed genes in the two inbred lines treated with exogenous MeJA. The BP enrichment of DEGs identified in the deep-sown inbred line Qi319 was mainly concentrated in cellular processes, followed by metabolic processes and biological regulation. CCs were mainly enriched in cells, followed by cell parts, organelles, and membranes. The main MFs enriched were catalytic activity, binding activity, and transport activity. The BPs enriched in DEGs from Zi330 were metabolic processes, cellular processes, and stress responses. In addition, the CC and MF terms with significant enrichment of DEGs in Zi330 sensitive inbred lines were basically the same as those of Qi319. However, Zi330 having more DEGs for the same enrichment terms indicated that exogenous MeJA mainly enhances the deep-sowing tolerance of maize by up-regulating or down-regulating more DEGs.

#### 2.2.4. Pathway Enrichment Analysis of Differentially Expressed Genes

In order to further understand the biological functions of differentially expressed genes, pathway enrichment analysis of differentially expressed genes of the two maize inbred lines under deep-sowing stress was conducted. The enrichment mainly involved cellular processes, environmental information processing, genetic information processing, metabolism, and organismal systems. Compared with the 3 cm control group, the significant enrichment pathways of the Qi319 inbred line under 15 cm deep-sowing stress mainly included transport and catabolism, global and overview maps, signal transduction, amino acid metabolism, energy metabolism, and environmental adaptation. The enrichment of the carbohydrate metabolism pathway was the most significant (Figure 5a). The significantly enriched pathways noted for the deep-sowing-sensitive inbred lines of Zi330 under deep-sowing stress included transport and catabolism, global and overview maps, amino acid metabolism, energy metabolism, carbohydrate metabolism, environmental adaptation, and other pathways; the enrichment of the translation pathway was the most significant (Figure 5b). In both inbred lines, multiple enrichment pathways were screened, including transport and catabolism, amino acid metabolism, energy metabolism, carbohydrate metabolism, signal transduction, transcription, and global and overview maps.

The pathway enrichment analysis of the differentially expressed genes treated with exogenous MeJA at the sowing depth of 15 cm mainly involved five branches: cellular processes, environmental information processing, genetic information processing, metabolism, and biological systems. Compared with the deep-sowing treatment, the significant enrichment pathways noted in the deep-sowing-tolerant Qi319 inbred line under exogenous MeJA treatment mainly included transport and catabolism, global and overview maps, signal transduction, amino acid metabolism, and carbohydrate metabolism. Among them, the enrichment degree of the translation pathway was the most significant (Figure 5c). The significant enrichment pathways noted in the deep-sowing-sensitive inbred line Zi330 included carbohydrate metabolism, amino acid metabolism, energy metabolism, metabolism of cofactors and vitamins, metabolism of terpenoids and polyketides, and nucleotide metabolism (Figure 5d). The pathways of carbohydrate metabolism, amino acid metabolism, global and overview maps, lipid metabolism, and signal transduction were simultaneously screened in the two inbred lines.

Compared with the normal sowing depth, the differential genes whose expression levels were up-regulated and down-regulated in different comparison groups reflected the molecular mechanism of mesocotyl response to deep-sowing stress in the two maize inbred lines, while the differential genes whose expression levels were opposite might reflect the reason for the deep-sowing tolerance of the maize inbred line Qi319. Taking deep-sowing stress as the control, it was speculated that the differential genes whose expression levels were up-regulated and down-regulated in different comparison groups under exogenous MeJA treatment might reflect the molecular mechanism of the mesocotyl response of the two maize inbred lines to exogenous hormones to alleviate deep-sowing stress. Compared with the control, 704 different genes were co-expressed in the two maize inbred lines with different tolerances to deep-sowing stress. Using |log_2_FC| ≥ 2 and *p* ≤ 0.05 as the criteria, 76 differentially expressed genes were identified. These significantly different genes are mainly involved in biological processes, such as plant–pathogen interaction, unsaturated fatty acid biosynthesis, fatty acid metabolism, regulation of photomorphogenesis, and regulation of intima systems (Figure 6a,b). Compared with the deep-sowing treatment, we used |log_2_FC| ≥ 2 and *p* ≤ 0.05 as the criteria to screen the differentially expressed genes in the two inbred lines under exogenous MeJA remission treatment and obtained 12 differentially expressed genes with significant co-expression. It can be seen from Figure 6 that these significantly different genes are mainly involved in arginine and proline metabolism, glutathione metabolism, terpene biosynthesis, mitochondrial calcium homeostasis, sesquiterpene biosynthesis, terpene compound biosynthesis, and other biological processes (Figure 6c,d).

Compared with the control, the differentially expressed genes that were up-regulated in the deep-sowing-tolerant inbred line Qi319 but down-regulated in the deep-sowing-sensitive inbred line Zi330 under deep-sowing stress were screened using |log_2_FC| ≥ 2 and *p* ≤ 0.05 as the criteria, and six significantly differentially expressed genes were obtained. From Table 3, it can be seen that these significantly different genes are mainly involved in biological processes such as plant hormone signal transduction, amino acid metabolism, alkaloid synthesis, transcriptional regulation, and amine metabolism.

#### 2.2.5. Validation of DEGs by qRT-PCR Analysis

qRT-PCR was used to verify 10 common DEGs of the two inbred lines under deep-sowing stress. The results are shown in Figure 7. The relative expression of the genes *Zm00001d029630*, *Zm00001d022517*, *Zm00001d018229*, *Zm00001d023859*, *Zm00001d017696*, *Zm00001d033071*, *Zm00001d039468, Zm00001d021089, Zm00001d008837*, and *Zm00001d052759* is consistent with the expression trend of RNA-seq sequencing results for the two inbred lines under deep-sowing stress. Based on the above analysis, the expression trends of the 10 genes were consistent with the results of RNA-seq sequencing, which fully demonstrated the reliability of our sequencing results.

#### 2.2.6. Gene Co-Expression Network Analysis

WGCNA is a method that can analyze gene expression patterns in multiple samples and identify gene sets (i.e., modules) with similar expression patterns. To identify co expression patterns among DEGs, we used WGCNA to cluster genes with similar expression patterns into modules. We constructed a co-expression network using the expression data of 24 sample genes with FPKM values >1 (Figure 8a). Hierarchical clustering tree analysis was used to identify co-expression modules via WGCNA. Each leaf on a clustering tree represents a gene, and a total of 15 co-expression modules were constructed. From the heat map of the gene co-expression network, it can be seen that the colors between the same modules are darker, while the colors between different modules are lighter. This indicated that the correlation between modules was small and that our gene co-expression module clustering results were reliable and could be further analyzed (Figure 8b). Then, we analyzed the relationship between modules and specific traits/phenotypes of plants at normal and deep sowing depths, under deep-sowing stress, and under exogenous hormone treatment. Previous studies have shown that the tolerance of maize to deep sowing is mainly related to mesocotyl elongation. Therefore, our interest was focused on analyzing modules with which mesocotyl elongation was significantly positively correlated using the royalblue and bisque4 modules (Figure 8c,d).

#### 2.2.7. Analysis of Hub Gene Interaction Networks in the Modules

To determine specific genes most likely to be important in deep-sowing tolerance, we additionally used the expression data to generate a co-expression network. The network was used to identify “hub genes” for treatment groups. This term refers to genes with high connectivity in a gene interaction network. In this study, the five genes with the highest kME values (feature gene connectivity) in the royalblue and bisque4 modules were classified as hub genes. The hub gene and its interacting genes were used to draw a gene co-expression network diagram (Figure 9). In the royalblue module (Figure 9a), the central gene, 103640916, is a member of the histone H2A family, which regulates gene expression and cell differentiation by interacting with DNA. The 103628268 is a member of the ubiquitin conjugating enzyme E2T (UBE2T) family, which mediates the ubiquitin–proteasome system and regulates gene expression. The 100217314, 103638413, and 100274062 are functionally unknown genes. In the bisque4 module (Figure 9b), the central gene, 103643355, is a recombinant protein of streptococcus pyogenes serotype M6rpsE, and the primary ribosomal protein S5 is crucial for the 30S ribosome biosynthesis of eukaryotic bacteria. The 103625782, as an effector, can interact with chloroplast proteins and regulate chloroplast signaling pathways, and it is a GLK target gene encoding enzymes in the chlorophyll biosynthesis pathway. GLKs are positive regulatory transcription factors that dominate chloroplast development in plant nuclei. The 100285290 is a FK506 binding protein (FKBPs) that plays different roles in many key processes of plant growth, development, and response to abiotic stress. The 542277 is a Z-type nucleic acid binding protein 1 that activates receptor interacting protein kinase 3, causing cell death during RNA and DNA viral infections. The 100283885 is a functionally unknown gene. Therefore, we speculate that transcriptional regulation, histones, ubiquitin protease, protein binding, and chlorophyll biosynthesis play important roles in maize deep-sowing tolerance and that the exogenous hormone MeJA plays a role in alleviating deep-sowing stress.

#### 2.2.8. Analysis of the Mechanism of Exogenous MeJA Alleviating Deep-Sowing Stress in Maize

Based on the experimental results, we established the following molecular models to explain the response mechanism to deep-sowing stress of two different inbred lines and the mechanism by which exogenous MeJA alleviates deep-sowing stress in maize (Figure 10).

## 3. Discussion

### 3.1. Phenotypic Analysis of Exogenous MeJA to Alleviate Deep-Sowing Stress

Drought is one of the major abiotic stresses. Increasing the sowing depth of maize can make maize use deep soil water efficiently so as to avoid the problem of maize yield reduction caused by drought due to the loss of soil surface water [12]. Studies have shown that within a certain range, with increasing sowing depth, the emergence time will be extended; the emergence rate will be reduced; MES, COL, and RL will be increased; and the overall degree of seedlings will be significantly deteriorated [6]. In this study, compared with the normal sowing depth, MES and MES + COL in the maize inbred lines Zi330 and Qi319 were significantly increased (*p* < 0.05), while SDL was significantly shortened under deep-sowing stress. In addition, compared with deep-sowing stress, MES, MES + COL, and SDL in the maize inbred lines Qi319 and Zi330 were significantly increased after the addition of exogenous MeJA with 1.5 μmol·L^−1^ under deep-sowing stress. This indicates that exogenous MeJA treatment of maize materials with different deep-sowing characteristics under deep-sowing stress can effectively alleviate the persecution of maize under deep-sowing stress.

### 3.2. Analysis of Relative Hormone Contents in Maize Mesocotyls under Exogenous MeJA Treatment

Studies have shown that under deep-sowing stress, the elongation of maize mesocotyls is largely regulated by various plant endogenous hormones, such as IAA, GA, CTK, ABA, and ETH [13,14]. The contents of GA_3_ and IAA in the mesocotyl of maize increased when GA_3_ and UCZ (the gibberellin synthesis inhibitor, tenobuzole) were applied at the same time under deep sowing, indicating that when exogenous GA_3_ was applied to alleviate deep-sowing stress, the mesocotyl of maize promoted the elongation of the mesocotyl by increasing the IAA content. IAA can acidify the mesocotyl epidermal cell wall and relax it, thereby regulating mesocotyl cell elongation [15]. GA_3_ promotes the elongation of mesocotyl and coleoptile [8], but its effect on mesocotyl elongation was not as great as that of ABA [16]. GA_3_ also increases the enhancement effect of ethylene and ABA on mesocotyl elongation [7]. CTK mainly promotes mesocotyl elongation by dividing rice mesocotyl cells [17]. In this study, after exogenous MeJA was added, IAA increased, while ABA and EBR decreased, which may have been due to the synergistic effect of EBR and IAA. The increase in the IAA content inhibited the synthesis of EBR and led to the decrease in its content. There was no significant difference in EBR content after exogenous MeJA was applied under deep-sowing stress, indicating that exogenous MeJA promoted the elongation of mesocotyls, mainly promoting the elongation or enlargement of cells under deep-sowing stress. Different from this study, Wu et al. [18] believed that the elongation of rice mesocotyls was mainly due to the promotion of cell division by ABA, and ABA and GA had a positive superposition effect on mesocotyl elongation. Therefore, whether the regulation mechanism of ABA with respect to mesocotyl elongation in different crops is the same needs to be further studied and demonstrated.

### 3.3. Cytological Analysis of Maize Mesocotyls under Exogenous MeJA Treatment

Mesocotyl elongation is cytologically divided into an increase in the number of cells and an increase in the volume of cells. Zhao et al. [19] showed that the mesocotyl of maize alleviated deep-sowing stress mainly by promoting an increase in mesocotyl cell length rather than the elongation of mesocotyls caused by the change in cell number, which was consistent with the results of this study. The results of this study showed that the cytological reason for promoting maize mesocotyls under deep-sowing conditions was mainly the increase in cell volume rather than the increase in cell number, and exogenous MeJA could promote cell elongation to increase mesocotyl length and enhance seedling exhumation ability. It was speculated that exogenous MeJA induced changes in different hormone contents in the mesocotyls of maize seedlings, thereby regulating cell growth. Moreover, there were differences in the relief effects of exogenous hormones between deep-sowing-tolerant inbred lines and deep-sowing-sensitive inbred lines. Therefore, in actual production, the particularity of materials should be fully considered in considering whether exogenous hormones should be added to alleviate deep-sowing stress.

### 3.4. Analysis of Key Differentially Expressed Genes under Deep-Sowing Stress and Exogenous MeJA Mitigation Treatment

The statistical analysis of differential genes showed that 704 differential genes were co-expressed in the mesocotyls of the two inbred lines under deep-sowing stress, and 276 differential genes were up-expressed in the deep-sowing-tolerant inbred line Qi319 but down-expressed in the deep-sowing-sensitive inbred line Qi330. Using |log_2_FC| ≥ 2 and *p* ≤ 0.05 as criteria, 76 differentially expressed genes with co-up-regulation were screened, and 6 differentially expressed genes with co-up-regulation were up-regulated in the deep-sowing-tolerant inbred line Qi319 but down-expressed in the deep-sowing-sensitive inbred line Zi330. These significant genes are mainly involved in unsaturated fat acid biosynthesis, fatty acid metabolism, endometrial system regulation, plant hormone signal transduction, amino acid metabolism, and other processes.

Lignin is a kind of aromaticity polymer, which is mainly involved in the secondary walls of plant cells. When the lignin content in plant tissue is too high, the cell elongation and growth of the tissue will be inhibited, mainly because of the low cell wall relaxation caused by the high lignin content [20]. Some studies also believe that peroxidase (POD) is induced by hydrogen peroxide (H_2_O_2_) and synthesizes too much lignin, which leads to the hardening of the cell wall and a reduction in relaxation, resulting in the inhibition of cell elongation and growth [21,22]. In this process, reducing nicotinamide adenine dinucleotide phosphate oxidase (NADPH), POD, and polyamine oxidase (PAO) are the three main sources of H_2_O_2_ [23]. Genes involved in the redox process, such as *GRMZM2G359298*, were up-regulated in both inbred lines after treatment with exogenous MeJA. It is suggested that peroxidase may be induced by exogenous MeJA, thereby eliminating the accumulation of H_2_O_2_ to reduce the oxidation and polymerization of monosaccharide alcohols in the cell wall into lignin and reducing the degree of cell elongation inhibition. These genes may be responsible for cell wall synthesis and cell elongation, suggesting that they may play an important role in mesocotyl elongation of maize under deep-sowing stress induced by MeJA.

Most terpenoids in plants belong to secondary metabolic compounds, which play a positive role in the growth and development of plants, physiological functions, and interactions between plants and environmental factors. Plant hormones such as gibberellin, abscisic acid, brassinolactone, and strigolactones (SLs) are also terpenoid derivatives [24]. At present, terpenoids are mainly studied in maize salt stress and resistance to disease, while few are studied under deep-sowing stress. In this study, transcriptome sequencing was performed on maize inbred lines with different deep-sowing-tolerance characteristics, and it was found that some genes were enriched in terpenoid biosynthesis under deep-sowing conditions when MeJA was applied. The synthesis of terpenoids may play an important role in alleviating the adaptability of maize under deep-sowing stress.

At present, the interaction between hormones related to the elongation of maize hypocotyl cells is still unclear. Zhao and Wang [25] believe that IAA mainly regulates the elongation of maize hypocotyls by increasing the synthesis and transportation of IAA and found that increasing IAA concentrations under deep-sowing conditions can effectively promote the elongation of maize mesocotyls. In addition, under the condition of deep sowing, auxin binding proteins actively participate in promoting the elongation of maize mesocotyls. Kutschera et al. [26] applied exogenous IAA treatment to maize mesocotyls and coleoptiles cultured in vitro and found that both tissues were significantly elongated after treatment. They also found that BR and IAA promoted cell elongation and growth by acidification and relaxation of the epidermal cell wall. Pan et al. [27] showed that exogenous GA_3_ also promoted the elongation of mesocotyls under deep-sowing conditions. In addition, the “auxin elongation hypothesis” has been confirmed and expanded. It is reported that auxin binding protein 1 and IAA receptor are involved in the maize auxin/light signal network [28]. Compared with the deep-sowing treatment, differential genes that were up-regulated in the hypocotyls of two maize inbred lines under exogenous MeJA treatment were screened using |log_2_FC| ≥ 2 and *p* ≤ 0.05 as the standards. Twelve differential genes were identified, and ten key genes, including the ornithine decarboxylase gene, the terpene synthase 7 gene, the ethylene response transcription factor 11 gene, the calcium one-way transporter 4 mitochondrial gene, the protein exordium gene, the peroxidase 64 gene, the adenine nucleotide α hydrolase-like superfamily protein gene, the glutamine decarboxylase 1 gene, the resistance protein rps2 gene, and the extensin-like protein gene, were preliminarily screened out, which laid the foundation for the functional identification and cloning of related genes in a later stage. Among them, the expression of the *LOC103632960* gene, which encodes ethylene response transcription factor 11, was up-regulated. Ethylene response factor (ERF) is an important transcription factor which is involved in the response of plants to ethylene and the regulation of plant growth and development. ERF family members widely exist in various tissues of plants and play an important role in histogenesis, flowering regulation, stress-resistance response, and other aspects [29]. Studies have shown that the overexpression of the tomato *terf2/leerf2* gene in tobacco and tomato regulates the induced expression of cold-resistance-related genes and enhances the cold resistance of transgenic plants [30]. The overexpression of the ptaerf194 gene in poplar improves the resistance of plants to drought stress by improving water-use efficiency and limiting water loss [31]. The expression of the *LOC10363309* gene, which encodes mitochondrial calcium one-way transporter 4, was up-regulated. The mitochondrial calcium one-way transporter MCU is a highly selective calcium channel located in the inner membrane of mitochondria, and it is the main medium for calcium ions to enter the mitochondrial matrix. Duan et al. [32] found that the mitochondrial calcium one-way transporter ghmcu4 negatively regulates cell elongation, inhibits *ghmcu4* gene expression, significantly increases ca^2+^ content, activates the calcium signaling pathway, further promotes H_2_O_2_ accumulation, and enhances plant physiological and biochemical metabolism, thereby promoting plant cell growth and fiber elongation. The expression of the *LOC107546763* gene, which encodes an extensin-like protein, was up-regulated. Extensin is the main structural protein in plant cell walls, and it is a glycoprotein rich in hydroxyproline. It exists widely in the cell walls of various plants, accounting for 1% to 15% of the primary wall dry weight in dicotyledons. It plays an important role in plant growth and development. Its activity affects plant morphogenesis and growth and development, and it plays an important role in cell wall formation, cell elongation, stress resistance, and other aspects [33]. In this study, the mesocotyl length, mesocotyl and coleoptile length, seedling length, and cell length of mesocotyl tissue of the two inbred lines increased after exogenous MeJA treatment, indicating that exogenous MeJA can promote cell elongation, increase the length of mesocotyls, and improve the ability of seedling emergence. Therefore, it is speculated that these genes can be induced to express after treatment with the exogenous hormone MeJA and may also be involved in the regulation of mesodermal elongation under deep-sowing stress.

## 4. Materials and Methods

### 4.1. Experimental Materials

According to the preliminary laboratory test basis, the deep-sowing-tolerant maize autotrophic line Qi319 and the deep-sowing-sensitive autotrophic line Zi330, which differed significantly in mesocotyl length under deep-sowing stress, were selected for deep-sowing stress and exogenous substances to alleviate deep-sowing stress. The exogenous substances were MeJA (Solarbio, Beijing, China).

### 4.2. Experimental Method

#### 4.2.1. Seedling Culture and Treatment

Plump, uniform, and undamaged maize seeds were selected, disinfected with 0.5% NaClO solution for 10 min, rinsed 3–5 times with distilled water, and soaked in distilled water for 12 h. Afterward, the water was blotted with sterilized filter paper, the seeds were sown in PVC tubes (17 cm in diameter and 50 cm in height) filled with sterilized vermiculite and distilled water in a ratio of 5 g: 1 mL, and the bottoms of the tubes were sealed with nylon meshes. Ten seeds were sown per PVC tube, and there were six replications per treatment. Seeds were covered with 3 cm of soil for the normal-sowing control and 15 cm of soil for the deep-sowing stress treatment. The seeds were cultivated in a dark incubator at a temperature of 25/20 °C during the day/night, and each PVC pipe was watered with 50 mL of the corresponding treatment solution every 2 days. Measurements of parameters related to deep-sowing tolerance were made 12 days after germination. Three biological replicates were set for each treatment [34].

According to the previous results of our research group, the sowing depth for deep-sowing stress was 15 cm, and the concentration of MeJA was 1.5 μmol·L^−1^. Experimental treatment: Treatment 1 (CK): 3 cm sowing depth + distilled water treatment; Treatment 2 (CM): 3 cm sowing depth +1.5 μmol·L^−1^ MeJA treatment; Treatment 3 (DS): 15 cm sowing depth + distilled water treatment; Treatment 4 (DM): 15 cm sowing depth +1.5 μmol·L^−1^ MeJA treatment.

#### 4.2.2. Measurements of Related Parameters of Deep Sowing

The measurement of phenotypic traits followed the method of Peng et al. [35]: mesocotyl length (MES), coleoptile length (COL), seedling length (SDL), root length (RL), seedling fresh weight (SDW), root fresh weight (RW), mesocotyl weight (MESW), and coleoptile weight (COLW) were measured with a scale and analytical balance after 12 days of seed germination. Ten plants were taken to calculate the average values. The paraffin section technique with safranin-fixed green staining was used to observe the histological structure of the middle parts of maize mesocotyls [36]. The extraction method for endogenous hormones was based on the method of Wang et al. [35]. The technique was repeated 3 times by liquid chromatography–mass spectrometry (HPLC-MS).

#### 4.2.3. Transcriptomic Analysis of Deep-Sowing Stress in Maize

After 12 days of seed germination, the vermiculite attached to the corn seedlings was washed, the residual water on the surface was sucked up, the mesocotyl was removed and placed in a 5 mL condensing tube, and the seeds were frozen in liquid nitrogen and stored at −80 °C.

RNA extraction, library construction, and sequencing: Total RNA was isolated and purified from 24 samples following the TRIzol (Invitrogen, Carlsbad, CA, USA) reagent procedure. An RNA-seq library was constructed with total RNA of each treated sample whose purity met the criteria for building the library, which was completed by BGI.

Quality assessment of sequencing results: Clean reads were obtained by filtering the raw reads obtained by sequencing, and subsequent analysis was based on clean reads. The filtering software SOAPnuke 1.X independently developed by BTU was used for filtering. The specific steps were as follows: (1) reads containing adapters (adapter contamination) were removed; (2) reads with N contents greater than 5% were removed; (3) low-quality reads were removed (reads with a quality score of less than 15 bases were defined as those with a total number of bases greater than 20%). The filtered clean reads were saved in FASTQ format. Subsequently, the clean reads for each sample were compared with the fourth version of the B73 maize reference genome using HISAT software (GCF_000005005.2_B73_RefGen_v4) [37]. RSEM software (1.3.1) was used to detect gene expression levels.

#### 4.2.4. Analysis of Differentially Expressed Genes

The method of Audic S et al. [38] was adopted to screen DEGs. DEGs were screened using |log_2_ fold change (FC)| ≥ 1 and FDR ≤ 0.001 as criteria among different comparison groups. WEGO software (v2.0) [39] was used to classify the functional notes of selected DEGs in the GO database. Based on KEGG pathway significant enrichment analysis (performed via the 49-year-old Kyoto Encyclopedia of Genes and Genomes public database: https://www.genome.jp/kegg/pathway.html, accessed on 5 March 2022), using hypergeometric inspection, the pathways with significant enrichment were identified. Pathways with Q values ≤ 0.05 were defined as those that were significantly enriched in DEGs. Significant enrichment of pathways can identify the most important biochemical metabolic pathways and signal transduction pathways involved in DEGs.

#### 4.2.5. Weighted Gene Co-Expression Network Analysis (WGCNA)

Using the gene expression data obtained by RNA-seq, the FPKM threshold was set to 1, the module similarity threshold was 0.25, the minimum number of genes in the module was 30, and the co-expressed gene module was constructed using the WGCNA R package. The co-expression modules were associated with phenotypic traits, and genes with kME values > 0.5 were selected as module members (kME is the characteristic gene connectivity for screening hub genes). Cytoscape_v3.9.1 software was used to visualize the gene interaction network of the core module. Finally, the hub genes in the core modules were determined according to the kME values and gene connectivity [40].

### 4.3. qRT-PCR of Differentially Expressed Genes

In order to ensure the accuracy and reliability of the experimental results, we used total RNA from each treatment material for library construction. The cDNA was reverse transcribed with an RNA simple total RNA Kit (Tiangen, Shanghai, China). Ten DEGs were randomly selected from two inbred lines under deep-sowing stress, and their specific primers were designed by Primer-BLAST on NCBI. qPCR amplification was performed via the quantum Studio 5 real-time PCR system (Thermo Fisher Scientific, Waltham, MA, USA) using super real premix plus (SYBR Green) (Tiangen, Shanghai, China). Each treatment had three technique replicates. In addition, each real-time PCR was performed at 20 µL. The reaction volume included 10 µL SuperReal PreMix Plus, 6 µL ddH_2_O, 0.8 µL forward primer (10 µmol/L), 0.8 µL reverse primer, 0.4 µL Rox reference dye, and 2 µL template cDNA. The amplification procedure was described with reference to Li et al. [41]. Using the actin gene as the internal reference gene, the gene expression level was analyzed using the 2^−∆∆^CT calculation method [42].

### 4.4. Statistics Analysis of Data

Statistical analysis and plotting of the data were conducted using Microsoft Excel 2019, one-way ANOVA was performed using IBM SPSS Statistics 21.0 software (*p* < 0.05), and slice cell-length measurements were performed using Image Pro Plus 6.0. The experimental results were presented as means ± standard errors (SEs).

## 5. Conclusions

Based on the above analysis, the mesocotyl lengths, mesocotyl and coleoptile lengths, and seedling lengths of the two inbred lines were significantly increased after treatment with exogenous MeJA under deep-sowing stress, indicating that exogenous MeJA can promote cell elongation, increase the length of the mesodermal axis, and improve the ability of seedling emergence. Compared with the deep-sowing treatment, exogenous MeJA alleviated the deep-sowing stress of maize mainly through transportation and catabolism, signal transduction, amino acid metabolism, lipid metabolism, translation, intimal system regulation, terpene biosynthesis, and other pathways. After the exogenous MeJA mitigation treatment, there were 12 significantly differentially expressed genes in the two inbred lines, and 10 key genes, such as the ornithine decarboxylase, terpene synthase 7, and ethylene response transcription factor 11 genes, were screened out. By regulating the elongation of hypocotyls under deep-sowing stress, the deep-sowing tolerance of maize was enhanced. Our research not only provides a new perspective for solving the problem of maize planting in arid and water-deficient areas, but also provides a reference for studying the mechanism of maize tolerance to deep sowing and alleviating maize deep-sowing stress with exogenous hormones.

## Figures and Tables

**Figure 1 ijms-25-10718-f001:**
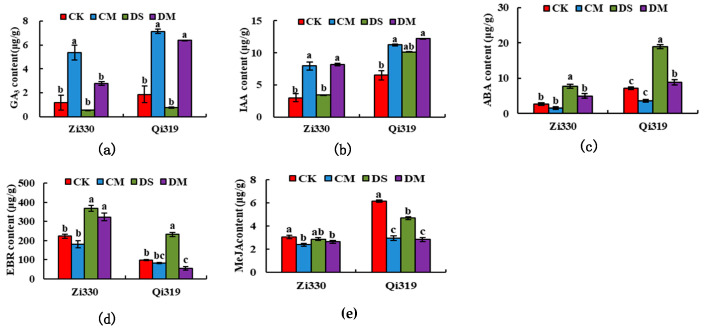
Effects of exogenous MeJA on endogenous hormones of maize inbred line seedlings under deep-sowing stress. CK: distilled water treatment at 3 cm sowing depth; CM: 3 cm sowing depth with 1.5 μmol·L^−1^ exogenous MeJA treatment; DS: distilled water treatment at 15 cm sowing depth; DM: 15 cm sowing depth with 1.5 μmol·L^−1^ exogenous MeJA treatment. Different lowercase letters represent the same inbred line with significant differences under different treatments (*p* < 0.05).

**Figure 2 ijms-25-10718-f002:**
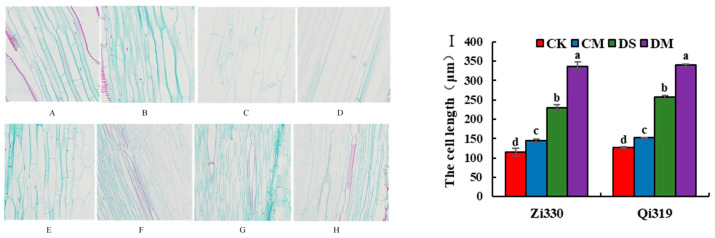
Longitudinal structure and cell length of mesocotyl cells of maize inbred lines treated with exogenous MeJA under deep-sowing stress. **A**: 3 cm sowing depth + distilled water treatment (Qi319); **B**: 3 cm sowing depth + MeJA treatment (Qi319); **C**: 15 cm sowing depth + distilled water treatment (Qi319); **D**: 15 cm sowing depth + MeJA treatment (Qi319); **E**: 3 cm sowing depth + distilled water treatment (Zi330); **F**: 3 cm seeding depth + MeJA treatment (Zi330); **G**: 15 cm sowing depth + distilled water treatment (Zi330); **H**: 15 cm seeding depth + MeJA treatment (Zi330); **I**: cell lengths of mesocotyls. CK: distilled water treatment at 3 cm sowing depth; CM: 3 cm sowing depth with 1.5 μmol·L^−1^ exogenous MeJA treatment; DS: distilled water treatment at 15 cm sowing depth; DM: 15 cm sowing depth with 1.5 μmol·L^−1^ exogenous MeJA treatment. Different lowercase letters represent the same inbred line with significant differences under different treatments (*p* < 0.05).

**Figure 3 ijms-25-10718-f003:**
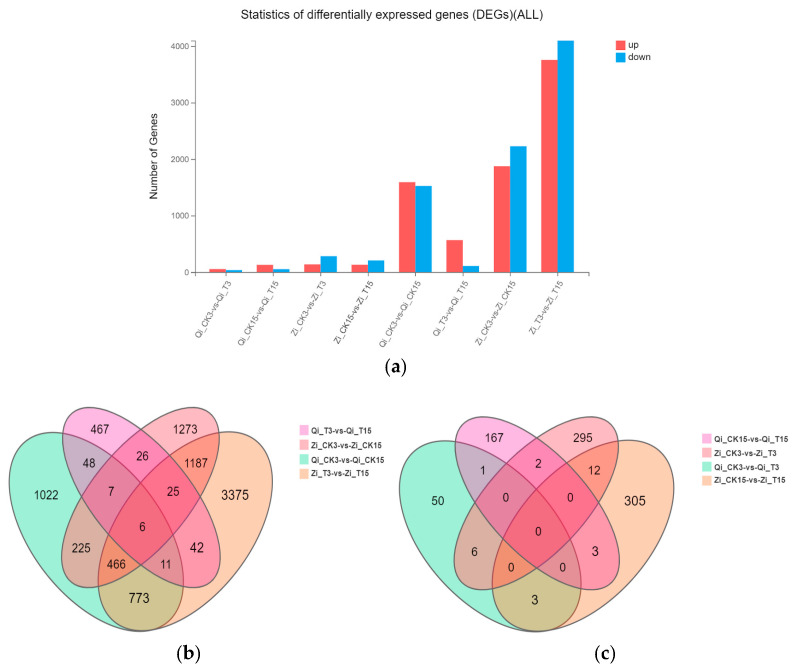
Differential gene expression analysis. (**a**) Number distribution of down-regulated DEGs in different comparison groups. (**b**,**c**) Venn diagram analysis of the normal sowing depth and deep-sowing stress and the normal sowing depth and deep-sowing stress with exogenous MeJA applied. Qi_CK3: Qi319 under 3 cm sowing depth and distilled water treatment; Qi_CK15: Qi319 distilled water treatment at 15 cm sowing depth; Qi_T3: Qi319 under 3 cm sowing depth and MeJA treatment; Qi_T15: Qi319 under 15 cm sowing depth and MeJA treatment; Zi_CK3: distilled water treatment of Zi330 at 3 cm sowing depth; Zi_CK15: distilled water treatment of Zi330 at 15 cm sowing depth; Zi_T3: MeJA treatment of Zi330 at 3 cm sowing depth; Zi_T15: MeJA treatment of Zi330 at 15 cm sowing depth.

**Figure 4 ijms-25-10718-f004:**
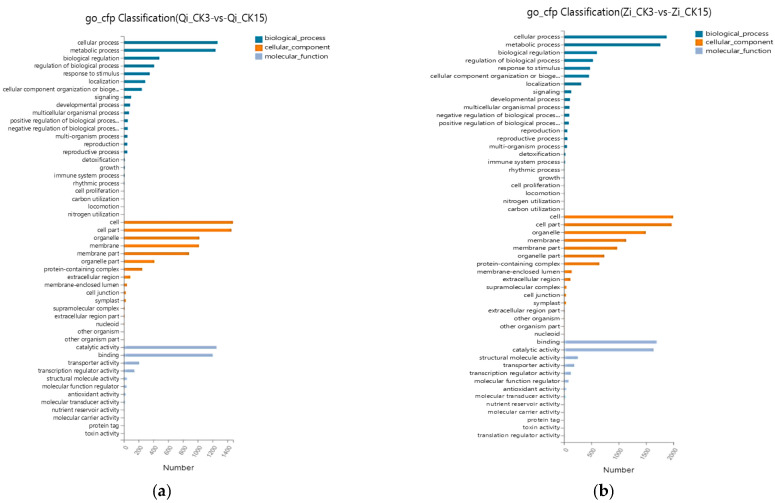
GO analysis of two inbred lines in different comparison groups. (**a**) GO analysis of Qi319 maize inbred line at normal sowing depth under deep-sowing stress. (**b**) GO analysis of Zi330 maize inbred line at normal sowing depth under deep-sowing stress. (**c**) GO analysis of Qi319 maize inbred line after adding exogenous MeJA under deep-sowing stress. (**d**) GO analysis of Zi330 maize inbred line after adding exogenous MeJA under deep-sowing stress. The treatments and abbreviations are the same as those given in Figure 1.

**Figure 5 ijms-25-10718-f005:**
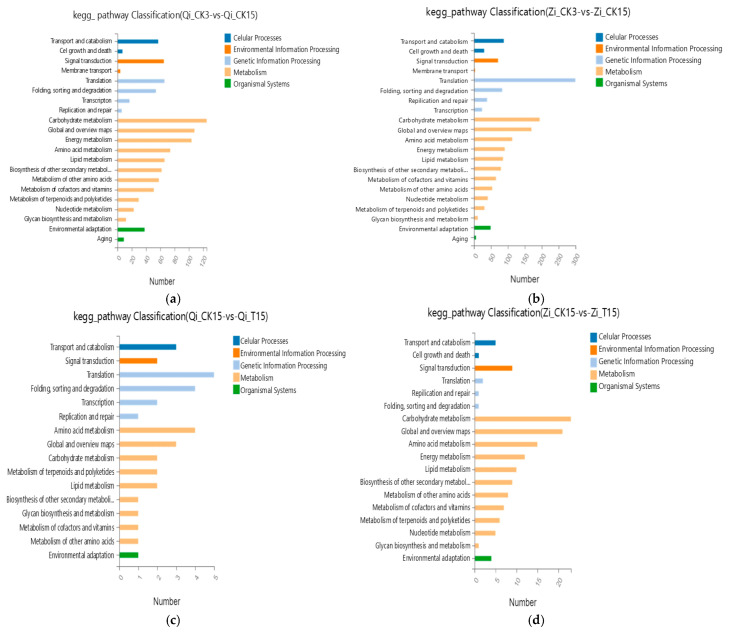
Pathway enrichment analysis of the two inbred lines in different comparison groups. (**a**) Pathway enrichment analysis of maize inbred line Qi319 at a normal sowing depth under deep-sowing stress. (**b**) Pathway enrichment analysis of maize inbred line Zi330 at a normal sowing depth under deep-sowing stress. (**c**) Pathway enrichment analysis of maize inbred line Qi319 after adding exogenous MeJA under deep-sowing stress. (**d**) Pathway enrichment analysis of maize inbred line Zi330 adding exogenous MeJA under deep-sowing stress. The treatments and abbreviations are the same as those given in Figure 1.

**Figure 6 ijms-25-10718-f006:**
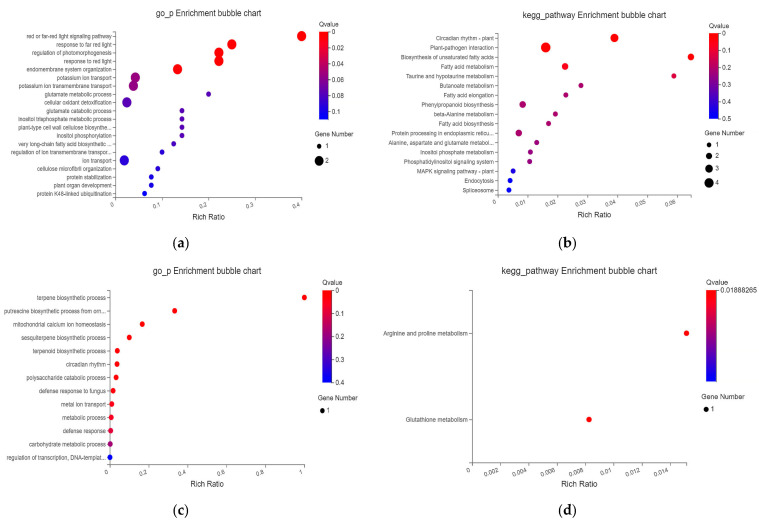
GO and KEGG analysis of expressed genes in different treatment groups. (**a**) GO analysis of two varieties under normal sowing and deep-sowing stress. (**b**) Pathway enrichment analysis of two cultivars under normal sowing and deep-sowing stress. (**c**) GO analysis of two varieties under deep-sowing stress after adding exogenous MeJA. (**d**) Pathway enrichment analysis of two cultivars under deep-sowing stress after adding exogenous MeJA.

**Figure 7 ijms-25-10718-f007:**
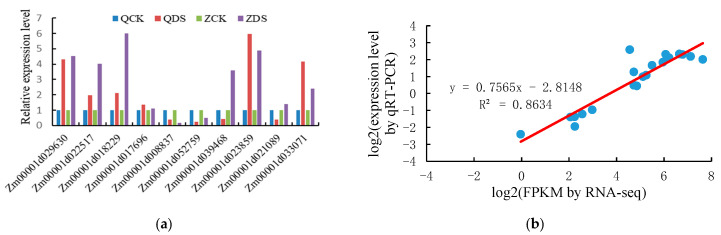
Real-time quantitative PCR validation of significantly up-regulated differentially expressed genes between two varieties under deep-sowing stress treatment. (**a**) The expression changes in response to the QCK, QDS, ZCK, and ZDS treatments for each candidate gene as measured by qRT-PCR. (**b**) Scatter plot showing the changes in the expression (log fold changes) of selected genes based on RNA-seq via qRT-PCR. The red line in the figure represents RNA seq, and the blue dots represent qRT-PCR. QCK: distilled water treatment of Qi319 at 3 cm sowing depth; QDS: distilled water treatment of Qi319 at 15 cm sowing depth; ZCK: distilled water treatment of Zi330 at 3 cm sowing depth; ZDS: distilled water treatment of Zi330 at 15 cm sowing depth.

**Figure 8 ijms-25-10718-f008:**
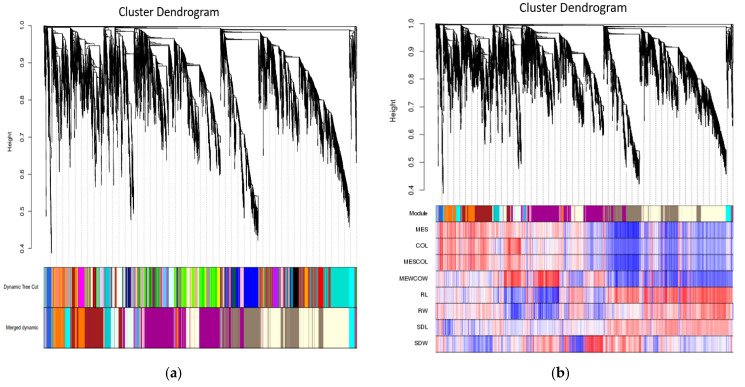
Gene cluster analysis and correlation analysis of phenotypes and modules. (**a**) Hierarchical clustering analysis of co-expression genes. Different colors represent all modules, with gray indicating genes that cannot be classified into any module by default. (**b**) Correlated heat maps between modules. A color block in the picture represents a numerical value. The redder the color, the higher the expression level, and the bluer the color, the lower the expression level. (**c**) Correlations between gene modules and phenotypes. Each tree diagram in the figure represents a module, each branch represents a gene, and the darker the color of each point (white → yellow → red), the stronger the connectivity between the two genes corresponding to the row and column. (**d**) Heat map of correlations between gene modules and phenotypes. The leftmost color block represents the module, and the rightmost color bar represents the correlation range. In the heatmap of the middle part, the darker the color, the higher the correlation, with red indicating positive correlation and blue indicating negative correlation. The numbers in each cell represent correlation and significance. MES: mesocotyl length; COL: coleoptile length; MESCOL: mesocotyl length and coleoptile length; MEWCOW: mesocotyl weight and coleoptile weight; RL: root length; RW: root fresh weight; SDL: seedling length; SDW: seedling fresh weight.

**Figure 9 ijms-25-10718-f009:**
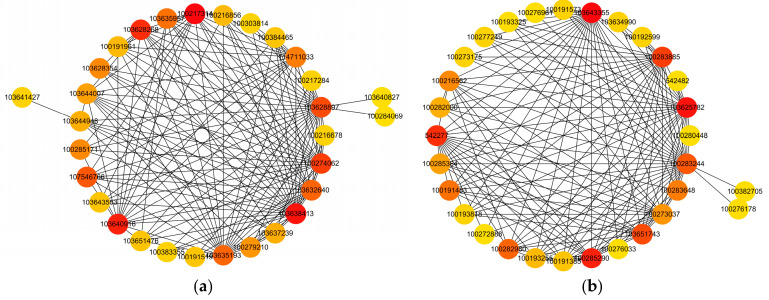
Analysis of hub gene network interaction in phenotypic significant enrichment modules. (**a**) Network interaction analysis of hub genes in royalblue module. (**b**) Network interaction analysis of hub genes in bisque4 module. The color gradients of the dots represent high or low soft thresholds of connectivity, with a redder dot color representing a higher soft threshold of connectivity. The color gradients of the dots represent high or low soft thresholds of connectivity, with a redder dot color representing a higher soft threshold of connectivity.

**Figure 10 ijms-25-10718-f010:**
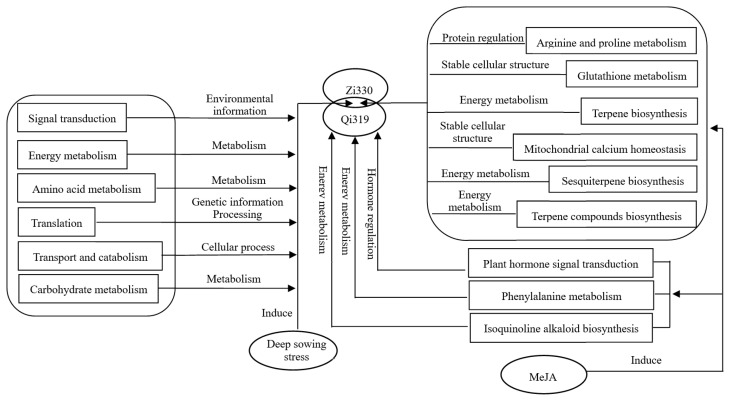
Model of the molecular mechanisms underlying deep-sowing tolerance and MeJA mitigation of deep-sowing-stress-induced damage in maize.

**Table 1 ijms-25-10718-t001:** The effect of exogenous MeJA treatment on seedling growth of maize inbred lines.

Material	Treatment	SDL (cm)	RL (cm)	SDW (g)	RW (g)
Zi330	CK	26.60 ± 0.81 c	33.10 ± 0.64 a	1.57 ± 0.06 a	1.38 ± 0.12 a
	CM	33.73 ± 1.18 a	29.00 ± 0.21 b	1.29 ± 0.02 b	1.20 ± 0.07 ab
	DS	29.23 ± 0.86 bc	20.27 ± 0.48 c	1.17 ± 0.04 b	0.96 ± 0.03 b
	DM	31.85 ± 1.55 ab	18.80 ± 0.30 c	1.26 ± 0.05 b	1.28 ± 0.01 a
Qi319	CK	34.10 ± 2.17 a	34.93 ± 0.64 a	1.21 ± 0.09 a	1.32 ± 0.03 a
	CM	27.47 ± 1.26 ab	34.67 ± 0.79 a	0.86 ± 0.04 b	1.33 ± 0.06 a
	DS	25.25 ± 5.05 b	23.40 ± 1.20 c	1.07 ± 0.10 ab	1.14 ± 0.07 b
	DM	33.90 ± 0.70 a	26.67 ± 0.64 b	1.15 ± 0.10 ab	1.34 ± 0.03 a

Different lowercase letters represent the same inbred line with significant differences under different treatments (*p* < 0.05). CK: distilled water treatment at 3 cm sowing depth; CM: 3 cm sowing depth with 1.5 μmol·L^−1^ exogenous MeJA treatment; DS: distilled water treatment at 15 cm sowing depth; DM: 15 cm sowing depth with 1.5 μmol·L^−1^ exogenous MeJA treatment. RL: root length; RW: root fresh weight; SDL: seedling length; SDW: seedling fresh weight.

**Table 2 ijms-25-10718-t002:** Growth characteristics of mesocotyls of maize inbred lines treated with exogenous MeJA.

Material	Treatment	MES (cm)	COL (cm)	MES + COL (cm)	MESW + COLW (g)
Zi330	CK	2.67 ± 0.17 c	2.57 ± 0.20 b	5.23 ± 0.35 c	0.33 ± 0.04 c
	CM	2.67 ± 0.33 c	2.23 ± 0.09 b	4.90 ± 0.25 c	0.25 ± 0.01 c
	DS	5.53 ± 0.29 b	6.30 ± 0.71 a	11.83 ± 0.43 a	0.75 ± 0.02 a
	DM	7.00 ± 0.60 a	2.80 ± 0.30 b	9.80 ± 0.30 b	0.62 ± 0.05 b
Qi319	CK	2.40 ± 0.21 d	3.20 ± 0.21 b	5.60 ± 0.21 c	0.24 ± 0.02 c
	CM	3.13 ± 0.09 c	2.50 ± 0.40 b	5.63 ± 0.32 c	0.23 ± 0.03 c
	DS	8.00 ± 0.20 b	3.53 ± 1.90 b	13.30 ± 1.00 b	0.37 ± 0.06 b
	DM	8.60 ± 0.12 a	7.10 ± 0.35 a	15.70 ± 0.26 a	0.52 ± 0.03 a

Different lowercase letters represent the same inbred line with significant differences under different treatments (*p* < 0.05). The treatments and abbreviations are the same as those given in Table 1. MES: mesocotyl length; COL: coleoptile length; MES + COL: mesocotyl length and coleoptile length; MESW + COLW: mesocotyl weight and coleoptile weight.

**Table 3 ijms-25-10718-t003:** Functional analysis of differential genes with different expression trends in the two varieties under deep-sowing treatment.

Gene ID	KEGG Pathway Desc	GO_*p* Desc
100272950	Plant hormone signal transduction	Transcription, DNA-templated; regulation of transcription, DNA-templated
100281647	NA	systemic acquired resistance
100284161	NA	NA
100284641	NA	Systemic acquired resistance
103638673	NA	NA
103641531	Glycine, serine, and threonine metabolism; tyrosine metabolism; phenylalanine metabolism; beta-alanine metabolism; isoquinoline alkaloid biosynthesis; tropane, piperidine, and pyridine alkaloid biosynthesis	Amine metabolism

## Data Availability

The datasets and materials used and/or analyzed during the current study are available from the corresponding author upon reasonable request.

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
