# Peer review of "Physiological Parameters and Transcriptomic Levels Reveal the Response Mechanism of Maize to Deep Sowing and the Mechanism of Exogenous MeJA to Alleviate Deep Sowing Stress"

_ijms, 2024, doi:10.3390/ijms251910718_

Round 1

Reviewer 1 Report

Comments and Suggestions for Authors

The paper addresses one of the most important aspects of wheat yields - the problem of drought. This problem does not only affect China, but also other grain-growing regions. The deep sowing seeds is one of the proposed solution. However, the work is generally described correctly but very workwise, not for independent reader. The some of the sentences in Results, MM, Discussion and conclusion should be more clearly and precise. As a result, it will not have the impact it should.

Figure 2. The anatomy photos should be the same orientation.

Table 3. This data should be in supplementary file or showed in more visible way.

Figure 3-6: The data are invisible.

Line 258: ?

Line 296 and 304:?

The material and methods: 4.1: You should precise your statement. It is incomprehensible.

How many seeds?

Line 557: Are the condition optimal? I don’t think so. It’s too hot.

The methods should be described more clear. Only the R package were used as the program to the data interpreting? Image Pro Plus 6.0. were used to the measurements or assessment? Precise. You should order and precise - what where why.

Line 637: MES, MESCOL and SDL, MeJA? It must be read the whole work to know what’s this. Precise.

The results laid a foundation for the candidate gene resources – is not clear.

Comments on the Quality of English Language

Generally correct.

Author Response

Comments 1: Figure 2. The anatomy photos should be the same orientation.

Response 1: Thank you for pointing this out. We agree with this comment. Therefore, we have adjusted the orientation of the anatomical photos.(Line 143)

Comments 2: Table 3. This data should be in supplementary file or showed in more visible way.

Response 2: Thank you for pointing this out. We have provided Table 3 as supplementary material, as shown in Table S1.

Comments 3: Figure 3-6: The data are invisible.

Response 3: Thank you for pointing this out. We agree with this comment. Therefore, we have adjusted the clarity of Figures 3-6. (Line187, Line 222, Line 259, Line 288)

Comments 4: Line 258: ?

Response 4: Thank you for pointing this out. We agree with this comment. We have modified this sentence. (Line 232- Line 235)

Comments 5: Line 296 and 304:?

Response 5: Thank you for pointing this out. We agree with this comment. We have modified this sentence. (Line 275- Line 280)

Comments 6: The material and methods: 4.1: You should precise your statement. It is incomprehensible.

Response 6: Thank you for pointing this out. We agree with this comment. We have modified the Materials and Methods 4.1, mainly introducing the two maize materials used in the experiment and the exogenous regulatory substances used.( Line529- Line533)

Comments 7: How many seeds?

Response 7: Thank you for pointing this out. Ten seeds were sown per PVC tube and 6 replications per treatment.( Line 541)

Comments 8: Line 557: Are the condition optimal? I don’t think so. It’s too hot.

Response 8: Thank you for pointing this out. Generally speaking, the germination temperature of maize seeds is between 15-30°C, with the most suitable temperature being 20-25°C. If the temperature is too high or too low, it will affect the germination rate and growth rate of corn seeds.

Comments 9: The methods should be described more clear. Only the R package were used as the program to the data interpreting? Image Pro Plus 6.0. were used to the measurements or assessment? Precise. You should order and precise - what where why.

Response 9: Thank you for pointing this out. We agree with this comment. We have modified the data analysis section. Meanwhile, I would like to clarify that Image Pro Plus 6.0 provides a variety of measurement and analysis tools, such as size measurement, shape analysis, cell counting, density analysis, and more. Users can use these tools to measure and analyze specific areas or objects in images to obtain more accurate and reliable data results. (Line 619- Line 622)

Comments 10: Line 637: MES, MESCOL and SDL, MeJA? It must be read the whole work to know what’s this. Precise.

Response 10: Thank you for pointing this out. We agree with this comment. In the article, we have modified MES, MESCOL, and SDL to mesocotyl length, mesocotyl and coleoptile length and seedling length. However, MeJA did not make any changes because the entire article was written in this way, and I am not sure if I want to modify it to Methyl jasmonate. (Line 624- Line 626)

Comments 11: The results laid a foundation for the candidate gene resources – is not clear.

Response 11: Agree. We have revised this sentence to“Our research not only provides a new perspective for solving the problem of maize planting in arid and water deficient areas, but also provides a reference for studying the mechanism of maize tolerance to deep sowing and alleviating maize deep sowing stress with exogenous hormones.”(Line 635- Line639)

Reviewer 2 Report

Comments and Suggestions for Authors

An interesting article. It concerns assessing the physiological and transcriptomic response of maize to the depth of sowing. The introduction refers to the conducted research. However, it lacks a research hypothesis, which should be added. The research methodology is correct. The authors presented the results in the form of graphs, tables and photos. The descriptions are correct, although not always legible. Graphs 8 and 9 are difficult to read, probably due to the large amount of data. The discussion is correct, supported by research by other authors. The conclusions correctly describe the obtained research results.

The work can be published in IJMS after minor corrections.

Comments on the Quality of English Language

English is legible.

Author Response

Comments 1: An interesting article. It concerns assessing the physiological and transcriptomic response of maize to the depth of sowing. The introduction refers to the conducted research. However, it lacks a research hypothesis, which should be added. The research methodology is correct. The authors presented the results in the form of graphs, tables and photos. The descriptions are correct, although not always legible. Graphs 8 and 9 are difficult to read, probably due to the large amount of data. The discussion is correct, supported by research by other authors. The conclusions correctly describe the obtained research results.

Response 1: Thank you for pointing this out. We agree with this comment. Therefore, we have proposed a hypothesis that exogenous application of MeJA can effectively alleviate the deep sowing stress of maize, promote the growth of the embryonic axis in maize seedlings, and improve their drought resistance. (Line 66- Line 68)

Reviewer 3 Report

Comments and Suggestions for Authors

The article with the title „Physiological and transcriptomic levels reveal the response mechanism of maize to deep sowing and the mechanism of ex-ogenous MeJA to alleviate deep sowing stress” has some important issues to be addressed.

The abstract is too long, please check the authors instructions - 200 words maximum. The abstract in this form is two times long as it should be.

The seedling length (SDL), root length (RL), seedling fresh weight (SDW) and root fresh weight (RW) are not indexes, two are morphological parameters and two are weights. Usually an index needs a formula to be determined. You only have obtained some values, please reformulate table 1 caption. Also in table footer list all abbreviation in order that you presented in the table.

Also the text writing could and should be improved, it seems that you took all tables in line and explained them literally. Write the most important results and only cite the tables or the figures not highlight the tables explanation as you did already.

As far as I know the figures caption should be standing alone, please see figure 2, so please write again the abbreviation do not force me to go and look for some abbreviations in figure 1.

Please do something with table 3, it is difficult to follow also why total raw reads has M abbreviation and total clean reads has the same abbreviation. Please change it, it is confusing.

Row 204. Please list the three differential genes because in my opinion it is an important result obtained. The figures are not comprehensive I barely can write some information from the figures, could you make the font bigger please? Or else delete them because the figures have no value for the readers in general. And make other comprehensive figures.

Something is wrong with figure 4. A and B from the left part of all the figures makes no sense. Please remove them. Also the fonts are different from the figures name and the clarity.

Figure 6 please improve figure caption and remove extra numbering A, B…and so on

In the material and methods section please add citation for all the methods used in the research.

Measurements of parameters not of indices.

The experimental methods look like a cocking recipe. ddH2O means double distillated? What ddH2O means and why did you abbreviate that way? Is it important for assuring the replicability of the research?

I also question what is the motivation of the study? Is this relevant in the field?

Figure 10 should be placed in the manuscript results section not in the conclusion section. Also, how was this model made? Something is wrong with it, please explain how this model could be trustful, it looks like it was made by hand in ppt.

99 % from the entire reference list is represented by Chinese authors. The scientific community aim for making the study international. If it is not of interest for the international scientific community perhaps it should be encouraged to be online elsewhere in my opinion.

Author Response

Comments 1: The abstract is too long, please check the authors instructions - 200 words maximum. The abstract in this form is two times long as it should be.

Response 1: Agree. We have revised the abstract to emphasize this point. (Line11- Line26)

Comments 2: The seedling length (SDL), root length (RL), seedling fresh weight (SDW) and root fresh weight (RW) are not indexes, two are morphological parameters and two are weights. Usually an index needs a formula to be determined. You only have obtained some values, please reformulate table 1 caption. Also in table footer list all abbreviation in order that you presented in the table.

Response 2: Thank you for pointing this out. We agree with this comment. Therefore, we have modified the title of Table 1. Meanwhile, all abbreviations displayed in the table are listed at the end of the table.( Line90, Line92- Line95)

Comments 3: Also the text writing could and should be improved, it seems that you took all tables in line and explained them literally. Write the most important results and only cite the tables or the figures not highlight the tables explanation as you did already.

Response 3: Thank you for pointing this out. We agree with this comment. Therefore, We have made modifications to the descriptions of Tables 2 and 3. ( Line81- Line89, Line96- Line101)

Comments 4: As far as I know the figures caption should be standing alone, please see figure 2, so please write again the abbreviation do not force me to go and look for some abbreviations in figure 1.

Response 4: Thank you for pointing this out. We agree with this comment. Therefore, we have added the abbreviation for Figure 2.( Line150- Line152 )

Comments 5: Please do something with table 3, it is difficult to follow also why total raw reads has M abbreviation and total clean reads has the same abbreviation. Please change it, it is confusing.

Response 5: Thank you for pointing this out. We agree with this comment. Therefore, we have modified the total raw reads and total clean reads in Table 3 by removing M. Meanwhile, Table 3 was uploaded as supplementary material.

Comments 6: Row 204. Please list the three differential genes because in my opinion it is an important result obtained. The figures are not comprehensive I barely can write some information from the figures, could you make the font bigger please? Or else delete them because the figures have no value for the readers in general. And make other comprehensive figures.

Response 6: Thank you for pointing this out. We agree with this comment. Therefore, we have listed three differentially expressed genes using Table S2 and adjusted the clarity of Figure 3.(Line 185, Line 187 )

Comments 7: Something is wrong with figure 4. A and B from the left part of all the figures makes no sense. Please remove them. Also the fonts are different from the figures name and the clarity.

Response 7: Thank you for pointing this out. We agree with this comment. Therefore, we have removed the letters A, B, C, and D from the image and made adjustments to the font and clarity. (Line187, Line 222, Line 259, Line 288)

Comments 8: Figure 6 please improve figure caption and remove extra numbering A, B…and so on

Response 8: Thank you for pointing this out. We agree with this comment. Therefore, we have improved the figure caption and removed the letters A, B, C, and D from the figure. (Line 288)

Comments 9: In the material and methods section please add citation for all the methods used in the research.

Response 9: Thank you for pointing this out. We agree with this comment. Therefore, We have added two references, 36 and 44. ( Line547, Line617)

Comments 10: Measurements of parameters not of indices.

Response 10: Thank you for pointing this out. We agree with this comment. Therefore, we have changed “indices” to “parameters”. ( Line554)

Comments 11: The experimental methods look like a cocking recipe. ddH2O means double distillated? What ddH2O means and why did you abbreviate that way? Is it important for assuring the replicability of the research?

Response 11: Agree. This should be distilled water. We are used to abbreviating it as ddH2O, we have changed ddH2O to distilled water.( Line537)

Comments 12: I also question what is the motivation of the study? Is this relevant in the field?

Response 12: Thank you for pointing this out. Deep sowing is one of the ways for maize to avoid drought and maintain moisture and seedlings. The elongation of the hypocotyl plays an important role in the emergence process of maize seeds from deep sowing. Research has shown that plant hormones have the effect of promoting elongation of the mesodermal axis. The aim of this study is to investigate the role of exogenous methyl jasmonate (MeJA) in maize hypocotyl growth and detect its regulatory network. The research results have deepened our understanding of plant hormone regulation of embryonic axis growth, which will help further explore and identify key genes in the plant hormone signaling regulatory network that affect embryonic axis development. This is our research motivation, perhaps not closely related to the field.

Comments 13: Figure 10 should be placed in the manuscript results section not in the conclusion section. Also, how was this model made? Something is wrong with it, please explain how this model could be trustful, it looks like it was made by hand in ppt.

Response 13: Thank you for pointing this out. This figure was summarized by the author based on the transcription results, and the author referred to other literature to create such a summary. So based on the experimental results of this article, we established the following molecular models to explain the response mechanisms of two different inbred lines to deep sowing stress and the mechanism of exogenous MeJA alleviating deep sowing stress in maize.

We have placed the Figure 10 in the results section, may I ask if it is appropriate? ( Line375- Line379)

Comments 14: 99% from the entire reference list is represented by Chinese authors. The scientific community aim for making the study international. If it is not of interest for the international scientific community perhaps it should be encouraged to be online elsewhere in my opinion.

Response 14: Thank you for pointing this out. Firstly, although some of the references are authored by Chinese authors, the articles are published in international journals, so they are also international. Secondly, we have also added some international references. (Line51- Line57, Line406- Line409)

Reviewer 4 Report

Comments and Suggestions for Authors

Comments are in the attached file

Author Response

Comments 1: I would suggest to change the title slightly. I would add “Physiological parameters and transcriptomic levels…

Response 1: Thank you for pointing this out. We agree with this comment. Therefore, we have changed the title to “Physiological parameters and transcriptomic levels…”. (Line2)

Comments 2: Line 44. Change is to are

Response 2: Thank you for pointing this out. We agree with this comment. Therefore, we have changed “is” to “are”. (Line32)

Comments 3: Lines 68-70. Sentence is a bit long and unclear in the end. Please rephrase it.

Response 3: Thank you for pointing this out. We agree with this comment. We have made modifications to this sentence.( Line61- Line62)

Comments 4: Ines 79-82. Sentence is too long and in the end, it does not say anything. It states “ To elucidate…” but it does not state in what way things will be elucidated.

Response 4: Thank you for pointing this out. We agree with this comment. We have made modifications to this sentence.( Line73- Line75)

Comments 5: Line 99. “When MeJa was treated with exogenous hormone..” Is this sentence correct. I Think something else should be instead of MeJa

Response 5: Thank you for pointing this out. We agree with this comment. we have changed the sentense to“When treated with deep sowing and exogenous MeJA,”.( Line82)

Comments 6: Line 116. hypocotile should be written without capital letter H

Response 6: Thank you for pointing this out. We agree with this comment. we have changed changed 'Hypocotile' to 'hypocotile'. ( Line96)

Comments 7: Line 177. Reads is written with capital first letter

Response 7: Thank you for pointing this out. We agree with this comment. we have changed changed ' Reads' to ' reads'. (Line158, Line159)

Comments 8: In the line 215 three main categories can have abbreviations written there and use them throughout the text. Then remove them from lines 219-221

Response 8: Thank you for pointing this out. We agree with this comment. we have made

revisions to the article based on your opinion. (Line196- Line201)

Comments 9: Line 256: ; instead of . Deep-seeded, seep seeding stress, deep sowing – can it be somehow a unified expression?

Response 9: Thank you for pointing this out. We agree with this comment. We have used a unified expression of words 'deep sowing '.( Line 174, Line 178, Line 199, Line 203, Line212, Line248, Line249, Line263, Line264)

Comments 10: Line 550. Plump, uniform and undamaged maize seeds were selected

Response 10: Thank you for pointing this out. We agree with this comment. We have revised the sentence 'Select plum, uniform and undamed maize seeds' to 'Plump, uniform and undamaged maize seeds were selected '. (Line536)

Comments 11: Line 572. 10 plants were taken to calculate the average value. Please use this style throughout (were taken, not we took)

Response 11: Thank you for pointing this out. We agree with this comment. We have revised the sentence ' Take 10 plants to calculate the average value' to '10 plants were taken to calculate the average value '. (Line559)

Comments 12: Line 634. Standard deviation or standard error? SD or SE in brackets

Response 12: Thank you for pointing this out. We have changed ' standard deviation ' to ' standard error '. ( Line622)

Round 2

Reviewer 1 Report

Comments and Suggestions for Authors

The manuscript is revised sufficiently.

Author Response

Comments 1: The manuscript is revised sufficiently.

Response 1: Thank you again for your significant suggestions.

Reviewer 3 Report

Comments and Suggestions for Authors

Row 546 measurement of the indices, you have replaced in all manuscript indices, why let it here?

Also you have not responded to the question about figure 10. How the model was done? Did you use a software; did you make it in ppt?

Author Response

Comments 1: Row 546 measurement of the indices, you have replaced in all manuscript indices, why let it here?

Response 1: Thank you for pointing this out. We agree with this comment. Therefore, we have changed “indices” to “parameters”. (Line 546)

Comments 2: Also you have not responded to the question about figure 10. How the model was done? Did you use a software; did you make it in ppt?

Response2: Thank you for pointing this out. The model in Figure 10 was drawn using drawing tools in Word without using any other software.